# A Review of Indoor Localization Methods Leveraging Smartphone Sensors and Spatial Context

**DOI:** 10.3390/s24216956

**Published:** 2024-10-30

**Authors:** Jiayi Li, Yinhao Song, Zhiliang Ma, Yu Liu, Cheng Chen

**Affiliations:** 1School of Civil Engineering, Tsinghua University, Beijing 100084, China; lijiayi22@mails.tsinghua.edu.cn (J.L.); y-liu20@tsinghua.edu.cn (Y.L.); 2Glodon Technology Co., Ltd., Beijing 100193, China; songyinhao@glodon.com; 3Department of Automation, Tsinghua University, Beijing 100084, China; chenc23@mails.tsinghua.edu.cn

**Keywords:** indoor localization, smartphone sensor, spatial context, data fusion

## Abstract

As Location-Based Services (LBSs) rapidly develop, indoor localization technology is garnering significant attention as a critical component. Smartphones have become tools for indoor localization due to their highly integrated sensors, fast-evolving computational capabilities, and widespread user adoption. With the rapid advancement of smartphones, methods for smartphone-based indoor localization have increasingly attracted attention. Although there are reviews on indoor localization, there is still a lack of systematic reviews focused on smartphone-based indoor localization methods. In particular, existing reviews have not systematically analyzed smartphone-based indoor localization methods or considered the combination of smartphone sensor data with prior knowledge of the indoor environment to enhance localization performance. In this study, through systematic retrieval and analysis, the existing research was first categorized into three types to dissect the strengths and weaknesses based on the types of data sources integrated, i.e., single sensor data sources, multi-sensor data fusion, and the combination of spatial context with sensor data. Then, four key issues are discussed and the research gaps in this field are summarized. Finally, a comprehensive conclusion is provided. This paper offers a systematic reference for research and technological applications related to smartphone-based indoor localization methods.

## 1. Introduction

Location-Based Services (LBSs) refers to various services provided based on the user’s geographic location [1]. These services are applicable in multiple domains, including navigation, social networking, targeted advertising, emergency evacuation, etc. Indoor localization plays a crucial role in LBSs because it is vital in areas such as smart parking, museum navigation, and large hospital guidance, significantly enhancing user experience and service efficiency. However, indoor localization is influenced by several factors, including the complexity of building layouts and topological relationships, the uncertainty of signal interference, the dynamic presence of people and objects, and the uneven distribution of temperature and humidity. These factors can significantly impact the accuracy, reliability, and real-time performance of localization [2]. To address the impact of these complex indoor environmental factors, various indoor localization methods have been proposed in existing research. These include geomagnetic localization [3], visual localization [4,5], Pedestrian Dead Reckoning (PDR) localization [6,7], barometer-based localization [8], inertial navigation systems, and methods based on Wi-Fi [9], Bluetooth [10], microphone [1], iBeacon [11], ultra-wideband [12], Zigbee [13], ultrasound [14], cellular networks [15], etc. The choice of indoor localization method depends on the specific indoor localization requirements, the indoor environment, and the characteristics of the selected localization terminal and its built-in sensors.

Indoor localization can be categorized into non-human localization and human localization. Non-human localization primarily focuses on devices such as robots [16] and drones [17], achieving precise localization through the integration of localization sensors into their hardware systems. Human localization, which is the focus of this study, typically relies on portable localization devices. Smartphones, as essential electronic products in our daily life, have become deeply integrated into various aspects of our everyday life. For instance, they are used for mobile payments, online food ordering and delivery tracking, and searching for nearby suppliers. The realization of these services relies on the localization capabilities of smartphones. In complex indoor environments such as large subway stations, underground parking lots, and underground complexes, their localization functions often encounter errors or even fail, so indoor localization methods need to be investigated specially.

Smartphones are becoming essential terminal devices for indoor localization due to their highly integrated sensors, rapidly evolving computational power, and widespread user adoption. As intelligent terminals for indoor localization, smartphones are equipped with a variety of built-in sensors. These sensors that can be utilized for indoor localization [18] include accelerometers [19], magnetometers [3], gyroscopes [20], light sensors [21], microphones [1], Bluetooth modules [10], Wi-Fi adapters [22], etc. These features highlight the advantages of sensor integration and portability. The variety and performance of smartphones sensors are typically influenced by factors such as the price, manufacturer, and production year. Not all smartphones come equipped with the full array of localization sensors. Basic sensors such as accelerometers, magnetometers, gyroscopes, Bluetooth modules, and Wi-Fi adapters are built into most budget smartphones. However, more advanced sensors, such as barometers and sophisticated light sensors, tend to be more closely tied to the price of the smartphone. Nonetheless, considering the rapid pace of smartphone advancements in recent years, the sensor type expansion and performance optimization is expected to accelerate significantly. Moreover, the enhanced computational capabilities of smartphones support the processing of complex data and the application of Artificial Intelligence (AI), which extends the functionality and application scenarios of smartphones in indoor localization, including through technologies such as voice recognition and image processing.

Currently, there are a number of review papers on indoor localization that analyze the research progress in this field from various perspectives. Some of these papers review indoor localization research related to robots [23] and drones [24], while others focus on specific indoor localization technologies such as visible light [25,26], ultra-wideband [27], Bluetooth [28], sound signals [29], Wi-Fi [30,31], RFID [32], etc. Moreover, some papers concentrate on the application of AI technologies like deep learning algorithms [33] and machine learning algorithms [34] in indoor localization. However, despite the broad application prospects of smartphone-based indoor localization methods, there is still a lack of systematic reviews in this area. Among the existing reviews, Subedi et al. [35] provided an overview of smartphone-based indoor localization systems based solely on Wi-Fi and Bluetooth wireless signals. Ashraf et al. [36] reviewed smartphone-based indoor localization methods that utilize Wi-Fi, Bluetooth, geomagnetics, inertial sensors, cameras, barometers, and the fusion of these sensor data. Naser et al. [37] explored various technologies and methods for smartphone-based indoor localization, analyzing factors such as sample size, walking patterns, phone posture, and the sensor types used in different studies. Gu et al. [38] reviewed research on the enhancement of indoor localization using spatial context but did not specifically focus on smartphone-related methods.

Given the current landscape, a systematic analysis of the critical challenges in smartphone-based indoor localization is essential. This includes an in-depth exploration of existing research methods, highlighting their strengths and weaknesses. Building on this foundation, it is crucial to review and organize the advancements, providing a comprehensive overview of the field’s research progress. Hence, a systematic search was conducted in the Web of Science, Scopus, and CNKI (China National Knowledge Infrastructure) databases for relevant papers published from 2015 onwards. It is important to explain that CNKI encompasses a wide range of resources including journals, master’s and doctoral dissertations, conference papers, yearbooks, statistical data, books, standards, and patents in China, and includes 99% of the academic journals published in China. In the Web of Science and Scopus databases, the keywords “Indoor Localization”, “Indoor positioning”, and “Smartphone” were searched, yielding a total of 2123 papers. After removing duplicate papers and excluding those with weaker relevance to the topic based on abstract review, 49 relevant papers were selected. In the CNKI database, 690 papers were retrieved. After abstract review and screening, 33 papers closely related to the research topic were selected. In total, 82 highly relevant papers were identified.

This paper reviews indoor localization methods that are based on smartphones according to the different types of data sources they rely on by using the selected papers. Section 2, Section 3 and Section 4 address indoor localization methods using single sensor data source, methods integrating multiple sensor data, and methods incorporating spatial context, respectively. Section 5 explores smartphone-based indoor localization methods from various aspects, i.e., a comparison of the three types of localization methods, the selection and optimization of indoor localization algorithms, AI techniques in smartphone-based indoor localization, and optimization strategies for key localization performance, and summarizes the research gaps in this field. The paper concludes with a summary. It provides valuable insights and references for researchers in related fields and offers theoretical support and practical guidance for the practical application of relevant technologies.

## 2. Indoor Localization Methods Using Single Sensor Data Source

Based on whether artificial external signal sources are required for localization, smartphone-based indoor localization methods using artificial single sensor data sources can be classified into two categories, i.e., the methods that do not rely on artificial external signal sources, and the methods that do.

Indoor localization methods using single sensor data sources that do not rely on artificial external signal sources operate independently of external infrastructure, relying instead on naturally occurring signals in the environment or pre-existing signals within the building for autonomous localization. Under these conditions, the types of built-in sensors in smartphones are relatively limited, such as magnetometers. To deal with the problem of geomagnetic fingerprints not being unique and similar fingerprints often being discontinuous, Song et al. [2] employed a fuzzy C-means clustering algorithm to cluster geomagnetic fingerprint data collected by smartphones. The research introduced location zone switching rules to constrain the matching areas and the localization process, and subsequently used the nearest neighbor algorithm for fingerprint matching. The results showed that the localization error was within 1.46 m, and the single-point matching time was within 7.95 ms. The limitation of this method is that the effectiveness of location region switching can be affected when the geomagnetic characteristics of the indoor environment are indistinct or unevenly distributed.

Indoor localization methods that rely on artificial external signal sources depend on infrastructure such as radio transmitters or base stations. In these methods, smartphones typically use a wider range of built-in sensors compared to those that do not rely on external signals. These sensors include Wi-Fi transceiver modules [9,39] Bluetooth modules [10], and cellular network modules (e.g., 4G, 5G), as well as microphones and camera sensors. Common smartphone-based indoor localization technologies that are based on radio waves include fingerprint matching [10], Round-Trip Time (RTT) [39], Time of Arrival (TOA), and Angle of Arrival (AOA). Guo et al. [39] chose Wi-Fi data for localization and proposed a new method to deal with the issue of inefficiency in existing Wi-Fi-based methods. This method uses both the RTT approach and RSS (Received Signal Strength)-based approach to estimate the location based on Wi-Fi data. The estimated results are input into a Scalar Kalman Filter (SKF) algorithm, which then performs data filtering, smoothing, and prediction to improve localization accuracy. The method was validated in a research center hall, achieving a localization accuracy within 1.435 m and a position update time of 0.19 s. This study suggests that further research into the methods that integrate smartphone inertial sensors is warranted because the wireless signal method is still facing challenges for continuous location. Zhao et al. [9] also utilized Wi-Fi data for localization. They proposed a method to address the device type differences among smartphones in Wi-Fi fingerprint matching. This method is based on domain adaptation theory from deep learning, treating smartphone device types as different domains. By using adversarial training to extract task-relevant and device-independent data features, and transferring the location information learned from the source domain to the target domain, accurate localization is achieved. Experiments were conducted in a corridor and a shopping mall, showing that this method provides better localization results. Further research is needed on the adaptability of this method to newer types of smartphones, characterized by extremely rapid updates in hardware and software. Unlike Refs. [9,39,40], Wi-Fi data are applied for 3D indoor localization. This method employs the Weighted Centroid (WC) algorithm for 2D plane localization and uses pedestrian activity recognition for coarse altitude estimation. It was validated in a floor area containing two rooms, achieving a 3D localization accuracy with a 2D plane error of 1.147 m and an altitude error of 0.305 m.

To better meet the requirements for a high-accuracy, low-cost, fast data update rate, and improved real-time performance, Lin et al. [1] utilized acoustic signals and the TOA approach for location estimation. During localization, the user’s smartphone speaker broadcasts an acoustic signal requesting location and anchor nodes receive the request and send the acoustic signal back to the smartphone microphone. The TOA approach is then used to estimate the distance between the smartphone and anchor nodes, thereby determining the location. The validation that was conducted in a single room showed that this system achieved high-precision real-time localization within 0.05 to 0.30 m, but without specific real-time performance data. Yu et al. [10] used Bluetooth signals for localization and proposed a method to deal with the problem of traditional methods only being able to represent a single-dimensional fingerprint point feature. During the offline phase, the Bluetooth module of the smartphone collects signal strength valued from four different orientations at each fingerprint point, which are used for feature extraction and Access Point (AP) weight assignment. In the online phase, the smartphone collects signal strength values and matches them with the smallest fuzzy distance using the K-Nearest Neighbors (KNN) algorithm, averaging these values to determine the final location estimation. This method was conducted in an office room, achieving an average localization accuracy within 1.61 m, and enables millisecond-level real-time localization.

Based on the previous analysis, localization methods using a single sensor data source are generally simpler, and are usually validated in simple environments. Due to their straightforward system architecture, these methods do not require complex data fusion algorithms, resulting in lower computational and power consumption demands on smartphones. However, single sensor data source-based methods are highly sensitive to environmental factors and are susceptible to variations in crowd density, changes in building structure, or electromagnetic interference, which can lead to unstable localization accuracy. Furthermore, if the data source experiences a failure, the entire localization system seems to become inoperative.

## 3. Indoor Localization Methods Using Multi-Sensor Data Fusion

This paper classifies smartphone-based indoor localization methods based on multi-sensor data fusion into two categories, i.e., “2D indoor localization methods” and “3D multi-floor indoor localization methods”.

### 3.1. Two-Dimensional Indoor Localization Methods

Multi-sensor data fusion in indoor localization involves various built-in smartphone sensors such as Wi-Fi data, geomagnetic data, Bluetooth data, and audio data. Different studies select and combine appropriate sensor data according to their needs to achieve complementary effects.

PDR localization uses inertial sensor data from smartphones, including accelerometers, magnetometers, and gyroscopes. Due to the independence from external signals, real-time capabilities, low cost, continuity, and potential for fusion with other sensor data, PDR localization has become a basic data source in the multi-sensor fusion localization method. To address the high cost of infrastructure installation for external signal sources, Kang et al. [41] proposed an optimized PDR called SmartPDR. This method uses inertial sensors to collect data, with gyroscope data used to compensate for magnetic field interference affecting the magnetometer. SmartPDR improves gait event detection accuracy by using Gaussian and moving average filters, and fusing gyroscope and magnetometer data for more accurate heading estimation. Step length is estimated using both static and dynamic models. The validation was conducted in a corridor of a hall, and the real-time localization accuracy was within 1.62 m. The limitation is that the algorithm complexity, power consumption, and the completeness of the dataset need to be improved. Wang et al. [7] proposed a “PDR+ Wi-Fi” method to address PDR cumulative errors and the discreteness of Wi-Fi localization by combining continuous state prediction from PDR localization with discrete measurement correction from Wi-Fi localization to enhance localization accuracy. In this method, a stacked neural network composed of recurrent neural networks and Long Short-Term Memory (LSTM) networks is used to recognize and remember Wi-Fi signal strength sequences. The validation was conducted in a laboratory room, achieving real-time location accuracy within 0.90 m. For future work, the approach that feeds the data from multiple sensors into the neural network for training and performs calculations during the actual localization process was mentioned.

Lu et al. [42] combined multiple data sources as “PDR+ Wi-Fi + geomagnetic” to fully utilize the advantages of each data source. First, the data were preprocessed using the Random Sample Consensus (RANSAC) algorithm to eliminate outliers. Then, the data were fused using an Adaptive Particle Filter (APF) to enhance localization accuracy, real-time performance, and robustness. Validation results indicated that the localization accuracy using PDR alone is 10.45 m. The “PDR+ Wi-Fi” method achieved an accuracy of 3.71 m, while the “PDR+ Wi-Fi+ geomagnetic” method further improved the accuracy to 1.02 m. Fusing more types of data sources can improve localization accuracy, but it will also increase the computational complexity and requirements for computing power. To address the challenges of low indoor localization accuracy, small signal coverage range, high cost of infrastructure, and poor system generalization capabilities, Chen et al. [43] introduced a “PDR + Audio” method. This method employs the Time Difference of Arrival (TDoA) technique for audio, utilizing audio data collected by smartphone microphones. The localization data obtained from the audio is then fused with the PDR results using a Kalman Filter (KF) algorithm. The validation was conducted in an office and an exhibition center hall, achieving real-time localization at 20 Hz with an accuracy within 0.23 m. Zhang et al. [44] designed a PDR localization method enhanced by user motion state recognition to resolve the insufficient adaptability of traditional PDR methods with different user motion states. It utilizes accelerometers and gyroscopes to collect gait and heading angle change data, which are then processed by a deep learning network, Wavelet–CNN (Convolutional Neural Networks), to recognize six complex user motion states, i.e., walking, jogging, jumping, standing, climbing stairs, and descending stairs. The recognized motion states are used to dynamically adjust step detection algorithms and step parameters, improving PDR accuracy. Experimental results showed that this method achieved a localization accuracy within 2.27 m, significantly better than traditional methods. It was found that conducting in-depth research into expanding more user motion states is essential for localization enhancement. In addition, deep learning methods perform well in user behavior recognition but require substantial training data and computational power [45,46]

In addition to “PDR+”, other methods of integration have also been implemented. Shu et al. [47] proposed a two-pass bidirectional Particle Filter (PF) method that fuses “Wi-Fi+ geomagnetic”, which uses the smartphone Wi-Fi transceiver module to detect signal strength for preliminary localization. Then, the backward PF algorithm is employed to estimate the initial position. Finally, the forward PF algorithm is used for fine-grained location estimation. In this approach, Wi-Fi data are used to adjust particle weights, while geomagnetic data are utilized for continuous position tracking. This method is useful for dealing with the problem of magnetic field disturbances and discontinuous Wi-Fi data. The validation was conducted in an office room, a parking lot, and a market. It showed that this method achieved indoor localization accuracy within a range of 3.5 m. Similarly, the "Wi-Fi+ geomagnetic" fusion method is also employed in Ref. [48]. The key difference in this research lies in the process: initial localization is first performed using geomagnetic fingerprint matching, and then the position estimate is refined by incorporating Wi-Fi data. This method was tested in various corridors and showed a 30% improvement in accuracy compared to other methods, though no specific accuracy data were provided. However, the localization computation time was reported to be within the range of 215 ms. In response to the issues of Bluetooth localization being prone to jitter and PDR cumulative errors, Feng et al. [45] proposed a “Wi-Fi+ BLE” method. This method utilizes the smartphone Bluetooth module to collect Bluetooth Low Energy (BLE) data and micro-inertial sensors for data collection. Then, an Extended Kalman Filter (EKF) algorithm is used to fuse the BLE fingerprint matching results with the PDR results. This method was validated in a floor of a building, achieving average localization accuracy within 1.17 m.

Based on the preceding analysis, due to the characteristics of smartphones and the built-in sensors, PDR localization is commonly used as a basic method for localization with smartphones [42,45]. Many multi-sensor data fusion methods for smartphone-based indoor localization use PDR results as the preliminary localization results, and then correct the PDR cumulative error by integrating data from sources such as Wi-Fi [6], geomagnetic fields [42], audio [43], and BLE [45] thereby improving the localization performance, including with regard to factors such as accuracy, real-time performance, etc.

### 3.2. Indoor 3D Localization Methods for Multi-Floor Buildings

Multi-floor indoor localization methods based on multi-sensor data fusion are primarily categorized into two types. The first type of floor-detection-based 3D indoor localization method divides the 3D space into two separate processing steps, i.e., vertical floor detection and 2D coordinate estimation. Either floor detection or 2D coordinate estimation can be performed first, depending on the requirements and constraints of the application. The second type of method treats the entire 3D space as a unified whole. The method involves constructing a comprehensive feature set for the space and performing 3D localization through feature matching.

In some methods, 2D coordinate estimation is performed first, followed by vertical floor detection. Chen et al. [8] utilized barometer and geomagnetic data for indoor 3D localization by using a smartphone to avoid the limitations on external signal infrastructure such as dead zones or weak coverage areas of wireless signals. Initially, geomagnetic data collected by the smartphone magnetometer is used for initial localization through fingerprint matching, followed by the application of the Dynamic Time Warping (DTW) algorithm to improve matching accuracy. Subsequently, a barometer is used to detect the user’s ascent and descent, providing more accurate floor detection information. This study was validated in a six-floor campus building, achieving an average localization accuracy within 2.00 m with an optimal geomagnetic data queue length of 2 s. The advantage of this 3D indoor localization method lies in the characteristics of geomagnetic and barometric sensors, which do not rely on external signal infrastructure. However, this method requires the phone to be parallel to the ground while collecting data.

A method for integrating fingerprinting and PDR localization has received focus, but space expanding from 2D to 3D is still deficient. Zhou et al. [49] used an EKF algorithm to integrate signal strength collected through a smartphone Bluetooth module with PDR results from inertial sensors. Vertical height change data collected from a barometer sensor were then used for floor detection. The validation was conducted in a two-story building, achieving 3D indoor localization with an accuracy of 1.03 m. Alternatively, floor detection could be performed before 2D coordinate localization, and this approach could also successfully achieve 3D indoor localization. Zhao et al. [20] introduced a method to deal with the sensitivity of Wi-Fi and barometer data to external environment changes, such as crowd density, humidity and temperature. During the offline phase, a fingerprint database containing information such as magnetic field intensity values and floor IDs is constructed. In the online phase, a support vector machine is first used to identify user activity patterns based on data from the smartphone inertial sensors. Based on user patterns, geomagnetic data are mapped to correct deviations in magnetometer data, enabling floor detection. Finally, geomagnetic fingerprint matching is used to estimate the 2D coordinate estimation, achieving indoor 3D localization.

In floor-detection-based 3D indoor localization, the accuracy of floor detection is crucial. To ensure the detection accuracy, the external signal infrastructure necessitates the deployment of more base stations or more comprehensive fingerprint databases, leading to high costs and complex operations. To deal with this problem, Wang et al. [50] used the smartphone Wi-Fi transceiver module to detect signal strength for monitoring user entry to obtain the initial position and then identified ascent and descent activities using a floor-change detection algorithm based on the “pressure–altitude” relationship to establish an estimation model for floor detection. This method achieved a floor detection accuracy of up to 85% by combining smartphone barometer data with a small amount of Wi-Fi data with rough barometer data. The barometer data for the start and end times for ascending and descending the stairs are crucial for accuracy improvement. Since absolute barometer data can be influenced by factors such as temperature, humidity, spatial enclosure, and different barometer sensors, Yi et al. [51] proposed a relative pressure map method, according to the pressure difference between the reference and target floors for floor detection, which is measured by smartphone barometer. This method has demonstrated 100% accuracy. In addition to using barometric data for floor detection, neural networks and other AI technologies can also be employed. Ai et al. [52] introduced machine learning algorithms, applying neural networks and the Weighted K-Nearest Neighbors (WKNN) algorithm for floor detection based on Wi-Fi fingerprint matching, achieving an accuracy of up to 99%.

Professional localization sensors for Unmanned Aerial Vehicles (UAVs) are too expensive. Adopting smartphones for the location modules of UAVs seems to be a solution. Liu et al. [53] proposed a smartphone-based 3D indoor localization method designed to assist in UAV localization. First, 3D coordinates of the quadcopter are obtained using the Wi-Fi RTT approach and an EKF algorithm is employed to address inaccuracies due to signal propagation errors. Then, using the smartphone inertial sensors and barometer data, the UAV pose changes are estimated. Finally, an Error-State KF algorithm integrates Wi-Fi data with pose change data, and an RTS smoothing algorithm is applied for location optimization. This method was validated in an office hall, achieving real-time localization accuracy within 1.36 m. The algorithms above are used for data fusion, and deep learning techniques in data fusion warrant further research to optimize data processing factors, such as feature extraction, data fusion, and signal propagation model learning [54,55].

Hence, regardless of the method used for indoor 3D localization, extending indoor localization from 2D to 3D requires the integration of a broader range of data sources. These data sources include barometric data, Wi-Fi data, and Bluetooth data, which assist in identifying ascent and descent patterns and performing floor detection, thereby enabling complex indoor 3D localization.

Based on the previous analysis, multi-sensor data fusion methods for smartphone-based indoor localization integrate the advantages of multi-sensor data, offering enhanced adaptability to complex indoor environments and potentially improving localization performance. When one sensor’s data are noisy or the sensor itself fails, data from other sensors can complement or correct it, ensuring the continuity and accuracy of indoor localization. Nevertheless, multi-sensor data fusion methods are generally more complex, requiring higher computational power and energy consumption from smartphones. Unlike 2D indoor localization methods, multi-floor 3D indoor localization requires additional data such as barometric data or other approaches to provide vertical displacement information, or needs to integrate more complex data processing techniques like K-means clustering and support vector machines to optimize the 3D localization performance.

## 4. Smartphone-Based Indoor Localization Methods Integrating Spatial Context with Sensor Data

Building spatial context [38] refers to a range of data types that can enhance indoor localization, such as map data, landmark data, image data, spatial models, grid data, and graph models. The data serve as prior information for smartphone-based indoor localization, providing environmental features and spatial constraints that help correct and optimize sensor-based localization, thereby improving the localization performance. Among them, sensor data can involve single-sensor data or multi-sensor data.

### 4.1. Methods for Integrating Map Data with Sensor Data

Map data play a crucial role in indoor localization by providing information on physical boundaries and obstacles, thus constraining navigable spaces to correct localization and align them with the actual environment [56]. As a significant type of data in the spatial context, map data offer effective spatial information for indoor localization. They can be represented in various forms, including grid maps, topological maps, vector maps, and 3D reality maps. Key information about the representation, acquisition methods, and integration with sensor data are summarized in Table 1.

Traditional wireless signal localization has blind spots, and PDR localization has cumulative errors. Zhou et al. [57] presented a method that utilizes a topological map composed of special points (e.g., corners, elevators, escalators, and stairs) to create an indoor navigation network to assist in PDR localization. This study collects data from the smartphone inertial sensors and barometer to detect user activities such as turning, using elevators, climbing stairs, and walking, and constructs a sequence of user activities. The user activity sequence is then matched with the indoor navigation network to estimate the location. Concurrently, data from the smartphone inertial sensor is used for PDR localization. Finally, a Hidden Markov Model (HMM) combines the user activity sequence matching result and PDR result by using training and decoding algorithms to infer the final position. This method was verified in an office building and a shopping mall, achieving a real-time localization error of 1.30 m. The limitation is that the method relies on the assumption that only labeled nodes can be considered. Hu et al. [58] proposed a two-layer feature-map-based indoor localization method using grid map representation to address the issues of low accuracy in Wi-Fi localization and poor stability in visual localization. Each grid in the map contains Channel State Information (CSI) fingerprint features, associated visual features, and location information. During the localization phase, images of the indoor environment are captured with the smartphone camera, and the Yolov5 algorithm is used to detect safety exit signs, initializing the HMM states with the visual localization results. CSI data collected through the smartphone Wi-Fi transceiver module are used to correct the image-based localization results. Finally, the forward algorithm integrates the two localization results. The method was verified in an office and a parking lot, achieving real-time localization with an accuracy within 1.04 m and 0.92 m, respectively. The single localization time was approximately 150 ms. The localization accuracy can be further reduced in locations with less signal interference and dense distribution of emergency exit signs.

Additionally, to enhance the localization performance in complex indoor environments, Han et al. [59] introduced a method that constrains PF algorithms with map data, represented as a grid map. The method initially completes PDR localization using the smartphone inertial sensor, then corrects the PDR cumulative error by integrating Wi-Fi and Bluetooth fingerprint matching results using a map-constrained PF approach. The main role of the map is to prevent particles from “passing through walls” by adjusting particle weights. Experiments in a university office building showed that this method achieved real-time localization with an accuracy within 0.89 m. In contrast, Wang et al. [60] not only integrated map data into the PF algorithm to avoid “passing through walls”, but also incorporated floor map topological information into the Wi-Fi fingerprint database construction to improve localization accuracy. The method was verified to achieve an average accuracy of 1.92 m in a school office building. Computational efficiency improvement of the fusion algorithm is a direction that warrants further research to better adapt smartphones. Likewise, Zhao et al. [61] used map data to correct PDR cumulative error; however, they integrated map data through a KF algorithm to correct gyroscope errors, further refining the PDR cumulative error. These studies collectively indicate that map data play a crucial role in enhancing localization performance.

**Table 1 sensors-24-06956-t001:** Key information related to map data.

Types of Spatial Context	References	Representation Forms of Map Data	Methods of Acquiring Map Data	Methods of Data Integration
Map data	[57]	Topological map	Not specified.	Use HMM to match pedestrian activity sequences with indoor road network nodes.
	[58]	Grid map	Add CSI (Channel State Information) data to the grid map and associate visual features with locations in the map in the form of feature descriptors.	Provide a reference framework for the localization system by dividing the indoor environment into grids, with each grid storing CSI fingerprints and visual features.
	[60]	Topological map	Perform sub-region segmentation and define topological relationships for the floor plan.	Topological information in the floor map was input into the algorithm in the form of adjacency relationships between reference points and GSSS values.
	[61]	Grid map	Extract channel areas from images and generate a binary image to represent walkable areas, providing a simplified map for localization.	Use a PF to project walking data onto a grid map. The particles represent parameters such as position, orientation, and scale, which were continuously updated based on the smartphone’s Inertial Measurement Unit (IMU) data.
	[62]	3D reality map	3D reconstruction.	Match features from the images captured by the smartphone with those in the real-world 3D map, and use techniques such as the Perspective-n-Point (PnP) algorithm to compute the spatial pose of the smartphone’s camera.

Map data used for smartphone-based indoor localization can be constructed manually, semi-automatically, or automatically. Kageyama et al. [62] described a manual method where images of informational boards on indoor walls were captured with a smartphone. These images were then manually processed to extract channel areas and generate a binary image representing passable areas, providing a simplified map for localization. Shoushtari et al. [63] outlined a semi-automatic approach to extract map data and a routing diagram from CAD drawings. This approach requires manual intervention to clean the data and perform geospatial analysis. Zhao et al. [64] proposed an automatic indoor map construction method. This method analyzes pedestrian trajectories and integrates Wi-Fi data patterns to identify spatial features such as stairs, doors, and corners, thereby determining the shape and dimensions of rooms and corridors. Clustering algorithms and the α-shape algorithm were used to distinguish between room and corridor trajectories, drawing the shapes of rooms, while clustering algorithms and Principal Component Analysis (PCA) identified the length and width of corridors. Manual methods are simple and direct but inefficient, suitable for small-scale or specific scenarios, and require dynamic updates; semi-automated methods balance efficiency and accuracy but still require manual intervention to ensure data quality; fully automated methods require complex calculations and model training to ensure high precision, and are an area that needs continuous optimization and improvement.

### 4.2. Localization Methods Combining Landmark Data and Sensor Data

Landmark data refer to elements within an environment that have significant physical features or distinctive indicators of user movement changes, which can be detected by visual sensors, inertial sensors, and so forth. Examples include doors, corners, and staircases. In addition, landmark data encompass spatial features formed by significant changes in radio waves or magnetic fields, such as points with the strongest Wi-Fi signals or locations with notable changes in magnetometer readings. Another form of landmark data includes manually created reference markers, such as QR codes. As a component of spatial context, landmark data provide recognizable spatial anchor points that enhance the accuracy and reliability of smartphone-based indoor localization. Key aspects of landmark data, including their representation, recognition methods, and integration with sensor data, are summarized in Table 2.

Landmarks characterized by significant physical features or noticeable changes in user movement status can provide localization references or correct localization errors for indoor localization. To correct PDR cumulative errors with landmarks, Zhou et al. [65] utilized accelerometer and gyroscope data from smartphones to collect user localization data within an indoor parking garage. A machine learning classifier is used to identify semantic environmental data such as speed bumps and turns as landmark data. The landmark data are then annotated on a map to construct an indoor road network topology map consisting of nodes and edges. Subsequently, a PF algorithm is employed to combine the indoor road network topology map to correct the PDR cumulative error. This method was validated in an underground parking lot, achieving real-time localization accuracy within an average range of 3.00 m.

In addition to identifying landmarks from sensor data for localization, behavior state landmarks can also be constructed. In response to the instability of Wi-Fi localization and PDR cumulative errors, He et al. [66] proposed a “PDR + Wi-Fi + Landmark” method.

Data are first collected separately through the smartphone Wi-Fi transceiver module, accelerometer, and gyroscope for Wi-Fi fingerprint matching and PDR localization. CNNs are then used to analyze accelerometer data to identify user behavior states, which are matched with recorded behavior states in a landmark database to correct errors in Wi-Fi fingerprint matching and PDR localization. During the offline phase of establishing the surface database, an improved K-means clustering algorithm is used to automatically identify landmarks. Validation showed that the method improved the average accuracy by 0.30 m compared to “PDR+ Wi-Fi” localization methods. As a similar study for tackling the instability of Wi-Fi localization and PDR cumulative errors, Ref. [67] differed in that it introduced an EKF algorithm to fuse Wi-Fi localization results, PDR results, and landmark matching results to determine the final localization result. Gu et al. [68] proposed a “PDR + Wi-Fi + Landmark + Map” method. Unlike the previous studies, this research established a landmark graph where landmarks served as nodes and passable paths as edges. The graph is a directed graph used to restrict user movement directions within a specific environment. For example, in a corridor environment, the user can only move in two directions, and the landmark graph provided constraints on movement in these directions. The above-mentioned papers propose that enhancements in user pattern recognition and landmark identification, respectively, are directions that warrant further exploration.

Signal features such as radio waves and magnetic field variations can also be used as landmark data. Due to PDR having a cumulative error, Zheng et al. [11] utilized iBeacon data as landmarks to assist in PDR localization. During the localization process, data from the smartphone inertial sensor are first used for PDR localization. Then, iBeacon data are collected through the smartphone Bluetooth module to determine whether the pedestrian had entered or exited a landmark area. A PF algorithm is used to fuse the PDR result, Bluetooth landmark data, and map data for the location. The validation was conducted in a corridor of a building. Comparative validation revealed that the “PDR + Conditional Landmark” method had a relatively higher average localization accuracy of 1.05 m.

QR codes are commonly used as landmarks for the localization of robots, but some studies have explored using QR codes as landmark data in smartphone-based indoor localization to help users estimate their location and orientation. Considering the high cost and complexity of external signal infrastructure, and the convenience of phone cameras for obtaining images, Refs. [69,70] adopted different methods for indoor localization. Ge et al. [69] designed a QR code landmark system for localization, along with a corresponding “coarse–fine” QR code landmark and detection algorithms. During the localization, the smartphone’s built-in camera sensor captures images of the environment containing “coarse–fine” QR code landmarks. The landmark detection algorithm is used to extract details from the images, including the QR code’s number from the encoding library and the QR code’s center position, orientation, and actual width. A mapping relationship between the “coarse–fine” QR codes and the world coordinate system is then established. Finally, the Efficient Perspective-n-Point (EPnP) algorithm is used to compute the camera’s center position and orientation. The experiment was conducted in a parking lot to validate that the location error was within 1.00 m and the orientation error 10°. Zhang et al. [70] developed a system based on the Gaode API. During localization, users first scan QR code landmarks on nearby walls using the smartphone, extracting location information and mapping it to Gaode web server for visualization. As users walk, PDR localization is used for location estimation, and when a QR code is scanned again, the PDR cumulative error will be corrected again. This process is repeated until the destination. The validation was conducted in a shopping mall, verifying the accuracy within 1.60 m.

### 4.3. Other Methods

Other spatial contexts include image data, spatial models, grid data, and graph models. Although these data types vary in form and processing methods, they play similar roles in spatial description and localization by providing rich spatial information to enhance indoor localization effectiveness. Smartphone-based indoor localization methods integrate image data and sensor data to achieve indoor localization through image matching and similarity computation, or by assisting in PDR localization [71] and Wi-Fi [72,73] in indoor localization.

Since visual-based indoor localization methods have issues of poor reliability [74], low computational efficiency [75], algorithm time consumption [5], large and complex spaces [5], etc., Liu et al. [74] proposed a method that integrates image matching, Wi-Fi fingerprint matching, and point cloud data matching. First, preliminary localization results are obtained through Wi-Fi fingerprint matching, which helps to limit the data range in the database. Then, the smartphone-captured image data and point cloud data are matched with the database data to obtain the smartphone position and orientation. This method was verified in a school office building, achieving real-time localization accuracy within 2.78 m. Limitations include insufficient accuracy, high computational complexity, the impact of coarse errors, and a lack of feature points. Zhang et al. [71] introduced a method that integrates image matching and PDR localization. After capturing images using a smartphone, a two-tier image-matching search strategy is used to correct the PDR cumulative error. This method was verified in a school office building, with an algorithm processing time of 0.42 s and localization accuracy within 0.93 m. Wang et al. [5] proposed a method based on grayscale histogram similarity computation. In the offline phase, images are captured with the smartphone, and grayscale histograms are extracted to establish a mapping between grayscale histograms and spatial locations, forming an image feature database. In the online phase, image similarity computation is performed and the weighted average method is used to obtain the user’s accurate location. This method was verified in a school office building, achieving a processing time of less than 1.7 s and an average accuracy within 0.30 m. Yang et al. [75] proposed a rapid visible light indoor localization system. During the localization, stripe images of encoded LED lights are captured with the smartphone camera, followed by binarization and stripe feature extraction. The extracted stripe regions are then decoded based on encoding rules to obtain the original information containing the LED light’s unique identifier, which is mapped to indoor space to locate. This method was verified in an office building, achieving real-time localization accuracy within 0.075 m. For better localization performance, it is necessary to attempt different methods, including increasing the types of features, introducing deep learning techniques, adding spatial constraints, setting phone parameters, etc., in visual-based methods.

Spatial models typically include building spatial information with 3D coordinate data. To realize multi-floor localization and implement 3D PF algorithms with floor transition, Jaworski et al. [76] used a spatial model to assist in sensor-based indoor localization. The spatial model is constructed through a web application, which includes information such as floor plans, transition areas, obstacle information, and the location of Wi-Fi APs. The method proposes a 3D PF algorithm to fuse spatial model data with Wi-Fi data, PDR results, and barometer data collected from the smartphone’s built-in sensors to achieve indoor localization. Validation showed that this method improves real-time localization accuracy in complex multi-layer indoor environments compared to traditional methods. Zhang et al. [4] proposed a spatial-model-assisted method. This spatial model referred to a spatial coordinate system for indoor localization and a conversion model from pixel coordinates to world coordinates related to visual localization. According to the method, images are collected by the smartphone’s built-in camera sensor to extract feature sequences and identify structural landmarks in the environment based on these sequences. Structural landmarks are then matched with the spatial model to correct the PDR cumulative error. The validation was conducted in a school office building, and this method achieved real-time localization accuracy within 0.025 m.

Grid data vary in content across different studies [19,77,78], typically involving geometric, topological, and semantic information about indoor spaces. Additionally, the role of grid data varies across studies. In Ref. [77], the grid model was mainly used to correct the PDR cumulative error. In Ref. [19], grid data were used to match user walking paths. In Ref. [78], a hybrid grid model was established for providing environmental constraints to aid and improve the effectiveness of Wi-Fi-based localization algorithms. The information of the hybrid grid model includes semantic data, topological data, and geometrical data.

The high complexity of the spatial environment can lead to errors in the PF results. Shang et al. [77] proposed a method to deal with this problem. This method performs dead reckoning using the smartphone’s inertial sensors and barometer, while the grid model provides a detailed and semantic representation of the indoor space. During localization, a backtracking grid filter is used to find backtracking points based on historical tracking data and topological maps, correcting the PDR cumulative error. This method was validated in a school office building, achieving real-time localization with average accuracy within 1.27 m. Storage load and computational complexity are key factors, and more information for the grid model is needed to improve location estimation. To alleviate influences on map-matching from trajectory estimation, pedestrian motion, and pedestrian indoor networks, Shi et al. [19] used the smartphone’s inertial sensors and barometer to obtain 3D location, velocity, and orientation data. According to the environmental features and pedestrian motion features recognized from the data, a Bi-directional LSTM network was used to detect the floor. Then, A grid search algorithm was employed for indoor grid matching. Finally, an error-ellipse-enhanced UKF algorithm was used to integrate sensor data, pedestrian movement, and indoor network information. The validation was conducted in a multi-floor building, achieving 3D indoor localization accuracy within 1.13 m. The limitation is that extracting features manually from data like Wi-Fi, barometer, and magnetometer data seems to be time-consuming, prone to omissions, and influenced by personal experience and judgment.

Graph models represent methods based on graphs. Chen et al. [79] proposed a method based on graph models to address the complexity of fingerprint localization and the lack of scalability of triangulation approaches. They generated a logical floor map based on Wi-Fi data characteristics and constructed a fingerprint map through network isomorphism problems. During the localization phase, Wi-Fi data collected from smartphones with specific applications are used in a Bayesian localization algorithm to achieve an average accuracy of 85% in real time. The main factor of the complexity of fingerprint localization is the uncertainty of RSS measurements, so Zheng et al. [80] constructed a graph-based model to capture the spatial relationships between Wi-Fi APs and reference points to assist in sensor-based indoor localization. The study demonstrated the effectiveness and robustness of this method through evaluation on a public indoor localization dataset. Wang et al. [81] used an infinite-weighted graph model to assist in localization for avoiding reliance on signal infrastructure. Data were collected through smartphone inertial sensors, geomagnetic sensors, and light sensors for PDR localization, geomagnetic fingerprint matching localization, and light fingerprint matching localization, respectively. A PF algorithm was used to fuse these results for final location estimates. The infinite-weighted graph model provides a reliable reference framework for particle initialization and long trajectory calibration in PDR localization. The validation was conducted in an office, a shopping mall, and a parking lot separately, with differences under different external conditions. It was demonstrated that this method achieved a real-time mean location error within 1.29–3.95 m.

Based on the previous analysis, the integration of spatial context with sensor data can significantly enhance the sensor-based indoor localization performance. Among the various types of spatial context, map data and landmark data serve as the primary sources to support localization. These data types exhibit relatively standardized forms and integration methods. However, other categories of spatial context still lack a unified data representation and fusion approach. While the incorporation of spatial context clearly improves localization accuracy and enables real-time localization, there is no experimental comparison that specifically analyzes the positive or negative impacts on real-time performance.

## 5. Discussion

In this section, we will discuss four key issues of smartphone-based indoor localization methods, i.e., a comparison of the three types of localization methods and selection and optimization of indoor localization algorithms, AI techniques in smartphone-based indoor localization, and optimization strategies for key localization performance. This section will also identify research gaps in this field.

### 5.1. Comparison of Three Types of Positioning Methods

Based on the previous analysis, both smartphone-based indoor localization methods relying on single sensor data sources, whether external or non-external, can support localization in simple indoor environments, with accuracy statistics ranging from 0.05 to 1.61 m. However, their disadvantage lies in the instability of single sensor data sources; if the data source experiences significant errors or failures, the localization is likely to fail.

In contrast, methods based on multi-sensor data fusion can leverage data from multiple sources to maintain continuity and accuracy in complex environments, particularly in multi-level 3D indoor localization. Statistical data indicate that the accuracy ranges from 0.23 to 3.5 m, with a wider error range compared to single sensor data source localization methods. This is primarily due to the validation environment extending from a single room to encompass floor levels or even multiple floors. The types of sensors are relatively limited due to the constraints of smartphone hardware integration. For complementary needs of various sensor data, there are fewer combinations of multi-sensors, with common configurations including the "PDR+" model, as shown in Table 3.

Additionally, multi-source data often require complex fusion algorithms for support, among which PF algorithms and KF algorithms, along with their improved versions, are the main data fusion algorithms. Most existing research has focused on modifying and optimizing these foundational algorithms for specific application scenarios.

Spatial context, as prior data, can expand the variety of data sources to a certain extent. It can provide rich environmental information for sensor-based localization to overcome the limitations of sensor types. Spatial context is highly diverse and flexible, and can be integrated with sensor data through various approaches, such as serving as spatial constraints for particle weighting, reducing the matching range of fingerprint data, and correcting cumulative errors in PDR. Statistical data show that the accuracy range is within 0.025–3.95 m. However, the efficient acquisition and automatic updating of data such as maps, landmarks, and grids within the spatial context still pose significant challenges that need to be resolved.

In indoor localization scenarios, suitable single sensor data source methods can be employed in simple layouts, such as single rooms or uncomplicated single-floor environments. However, in single-floor scenarios with numerous obstacles, high foot traffic, and dynamic layouts, it is necessary to choose and optimize methods from multiple perspectives, including data fusion, algorithm design, and cost control. The key distinction in multi-floor indoor localization methods is the need to consider vertical position changes. Barometers are commonly used as auxiliary sensors for vertical location estimation. However, barometers can be influenced by external factors such as temperature, humidity, and air circulation, especially in large multi-floor underground environments, where their effectiveness requires further validation.

### 5.2. Selection and Optimization of Smartphone-Based IndoorLocalization Algorithms

Smartphone-based indoor localization methods encompass a wide range of algorithms with flexible applications. These algorithms include multi-modal data fusion algorithms, Wi-Fi fingerprinting algorithms, Bluetooth fingerprinting algorithms, geomagnetic fingerprinting algorithms, floor localization algorithms, map matching algorithms, user behavior recognition algorithms, and landmark identification and update algorithms, as well as others. When selecting a single-source indoor localization algorithm, it is crucial to consider factors such as the characteristics of the data source, indoor environmental conditions, and the desired localization accuracy. Multi-modal data fusion algorithms for indoor localization involve two or more data sources, and the selection and optimization of such algorithms require careful consideration of the processing approach for each type of data and the combination of all multi-modal data.

Different combinations of multimodal data, such as “PDR+ Wi-Fi” and “Barometer+ geomagnetism”, often require different fusion approaches. For instance, fusing Wi-Fi fingerprinting results with PDR results can be achieved using a PF algorithm [7]. Meanwhile, integrating geomagnetic fingerprinting, Wi-Fi fingerprinting, and PDR results may require APF combined with the RANSAC algorithm [42]. Similarly, combining BLE fingerprinting with PDR results can be conducted using a KF [45]. Although utilizing the same data source combination, the specific multi-modal data fusion algorithm can vary. For example, the PDR method, which combines data from accelerometers, magnetometers, and gyroscopes, can either be enhanced by deep learning-based human activity recognition to assist in localization [44], or it can rely on traditional methods like gait event detection, heading estimation, and step length estimation for PDR localization [41]. For the "Wi-Fi + Geomagnetic" combination, localization can be performed independently using geomagnetic and Wi-Fi data, followed by bidirectional particle filtering to fuse the two results [47]. Alternatively, geomagnetic fingerprinting results can be corrected using Wi-Fi data to reduce localization errors [48].

### 5.3. AI Techniques in Smartphone-Based Indoor Localization

Machine learning and deep learning are two prevalent categories of AI techniques applied in smartphone-based indoor localization. The key components of AI techniques involved in smartphone-based indoor localization methods include algorithms or models, utilization, and datasets.

Machine learning algorithms or models typically rely on traditional mathematical models, such as K-means clustering and decision trees. These algorithms or models depend on manually engineered features and are suitable for handling relatively simple tasks, such as classification, regression, and clustering, as shown in Table 4. They are widely applied in Wi-Fi fingerprint matching, landmark classification, and user behavior pattern recognition. Due to the relatively simple structure of machine learning algorithms or models, their required dataset sizes are moderate and their computational resource demands are low, making them suitable for real-time execution on resource-constrained smartphones. However, manual feature engineering often relies on the knowledge and expertise of domain specialists to design appropriate features, which may introduce subjectivity and limitations in feature selection. Moreover, manual feature extraction may fail to adequately capture the complex patterns and relationships within the data, particularly when dealing with high-dimensional data, potentially resulting in suboptimal model performance.

Deep learning leverages multi-layer neural networks, such as CNN and LSTM, to automatically extract features from large-scale datasets. This approach is particularly effective for handling complex high-dimensional data, such as images, videos, and time-series data, enabling the processing of indoor localization tasks in intricate scenarios, including user behavior recognition, Wi-Fi signal temporal analysis, and visual data processing, as shown in Table 5. However, the multi-layer structure of deep learning algorithms or models results in a vast number of parameters and requires extensive data for training, placing higher demands on the computational power, storage capacity, and battery life of smartphones. Additionally, the training process for deep learning is lengthy, and the optimization process is complex, further complicating its implementation in practical applications.

### 5.4. Optimization Strategies for Key Localization Performance

As shown in Table 5, accuracy and real-time performance are the two core indicators for localization performance evaluation. The methods reviewed in this paper demonstrate accuracy ranging from 0.025 to 3.95 m, with most falling between 1 and 2 m. Given the strong human–device interaction characteristics of smartphone-based indoor localization, this accuracy range remains acceptable.

Enhancing real-time performance is a more critical focus. As shown in Table 4, existing research has explored indoor localization methods within the computational capabilities of smartphones and has managed to meet real-time localization requirements. However, as the complexity of fused data increases or the volume of data grows, real-time performance analysis and the validation of localization methods become increasingly crucial.

(1)Real-time performance-influencing factors and evaluation indicators

Key factors affecting real-time performance include algorithm complexity, the number of particles in PF algorithms, and fingerprint matching numbers. Real-time performance metrics include computational efficiency, computation time, single-point matching duration, single-localization time, and position update rate. Reducing redundant data through map constraints and other methods can enhance computation speed. KF and PF algorithms are commonly used in real-time systems and perform well in terms of real-time performance, making them priority choices for algorithm improvements.

(2)Ways to improve real-time performance

To further enhance the real-time localization performance, the optimization of existing algorithms is necessary. This could involve simplifying algorithms to reduce computational complexity, introducing constraints into traditional algorithms to decrease the computational load, or incorporating technologies such as cloud computing or edge computing. These optimization strategies can help improve the real-time capability and reliability of localization methods, thereby meeting the demands of more complex application scenarios for smartphone-based indoor localization.

### 5.5. Research Gaps

As the complexity of algorithms in smartphone-based indoor localization methods increases, a key area for further research is how to design and optimize algorithms to enhance localization performance, particularly in scenarios where smartphone computational resources are limited. The challenge lies in developing low-complexity, high-efficiency algorithms that still deliver accurate results. Additionally, leveraging AI techniques to achieve high-precision real-time localization is another critical focus. For instance, machine learning algorithms or deep learning algorithms can be employed to accurately identify different motion states, such as walking or being in a vehicle, which would allow for adjustments to the structure of PF algorithms. This would enable the selective use of various data sources and adaptive updating of particle weights, ultimately improving overall localization accuracy. See Table 6.

(1)Lack of a unified spatial context application strategy for assisting in sensor-based localization

In smartphone-based indoor localization methods that integrate spatial context with sensor data, the lack of a unified data application strategy is influenced by differences in the format, content, detail, and organization of spatial context across various buildings. It is necessary to establish application models for spatial context in indoor localization and to explore multi-modal data fusion localization methods based on this context. Spatial context data, such as maps and landmarks, are often acquired and updated through manual or semi-automatic methods. Achieving efficient and automated acquisition and updating these data remains an area for further research. Data could be crowdsourced by using smartphone users’ movement trajectories within buildings to automatically generate floor plans and landmarks. Alternatively, deep learning algorithms could be employed to automatically identify and annotate important landmarks.

(2)Rational calculation strategy to support real-time performance under complex data and algorithms

Smartphones can provide powerful support for indoor localization, but continuously running localization algorithms and handling large volumes of data pose significant challenges to the device’s battery life and computational capacity. Enhancing the real-time performance of localization methods on computationally constrained smartphones remains a critical issue. For example, migrating computationally intensive tasks, such as deep learning model training and large-scale data analysis, to cloud or edge devices can alleviate the computational burden on smartphones, allowing them to focus on data collection and basic processing. This approach can significantly improve the responsiveness and real-time performance of localization. Additionally, leveraging AI accelerators on cloud or edge devices (such as Google Edge TPU or NVIDIA Jetson) can expedite model inference processes and reduce computational latency on the smartphone.

## 6. Conclusions

This study reviews smartphone-based indoor localization methods by systematically searching and analyzing the literature. It classifies and discusses various methods, providing an in-depth exploration of the strengths and weaknesses of each method. The following summarizes the findings.

(1)The single-sensor localization method can achieve relatively accurate localization in specific environments with low cost and low complexity. Nevertheless, due to the inherent limitations of single sensors, they struggle to meet localization requirements in complex indoor environments.(2)The multi-sensor data fusion method can significantly improve the accuracy and robustness of indoor localization by integrating sensor data with complementary characteristics. Compared to localization in a 2D localization, multi-floor localization in a 3D space typically requires the integration of specialized sensor data or the employment of more complex methods to fuse multi-source localization data. In both 2D and 3D localization scenarios, multi-sensor fusion demonstrates remarkable advantages in enhancing accuracy and reliability. Nonetheless, multi-sensor fusion algorithms are relatively complex, and the data processing demands high computational power and energy consumption, posing challenges for the application of smartphones in indoor localization.(3)In the methods that integrate spatial context and sensor data, elements such as maps, landmarks, image data, spatial models, grid data, and graph models play a crucial role. These elements enhance positioning accuracy by providing physical boundaries, obstacle information, visual features, and structured spatial representations, which help correct errors and improve the effectiveness of PF algorithms. However, compared to sensor data, different modalities of architectural spatial context vary in format, content, level of detail, and organization. Therefore, it is essential to explore and innovate application models and methods for integrating architectural spatial context in indoor localization.

The primary contribution of this study is a systematic analysis of smartphone-based indoor localization methods, which organizes the latest research findings and provides valuable theoretical references and guidance for academic research and technical applications. This study summarizes the strengths and weaknesses of existing methods and, based on the limitations and challenges present in current research, proposes key areas and directions for future studies. However, the study also has some limitations. Due to space and resource constraints, the literature review only covers the Web of Science, Scopus, and CNKI databases, and therefore may have excluded some relevant research from other databases. Future research should consider broader database searches to ensure the comprehensiveness and representativeness of the results.

## Figures and Tables

**Table 2 sensors-24-06956-t002:** Key information related to landmarks.

Types of Spatial Context	Reference	Representation Forms of Landmarks	Landmark Recognition Methods	Methods of Data Integration
Landmark data	[65]	Specific locations or features, such as speed bumps and so on	Machine learning methods	Landmarks were used to correct the cumulative error of PDR; the weights of particles were updated based on the recognition results of environmental landmarks
	[66]	Clustered landmarks, such as specific indoor areas, room entrances and so on	Cluster analysis	Landmarks were used to identify the user’s behavior state and corresponding coordinates for position estimation correction
	[67]	Specific environmental features, such as elevators, staircases, and so on.	Decision tree methods	Landmark data and sensor data were input into the EKF for data integration
	[68]	Specific points or objects, such as doorways, staircases, and so on	Obtains landmarks from floor plans and identifies them from smartphone sensor data	Landmarks were used to construct a landmark map to assist heading estimation in the KF and were also used to calibrate the cumulative error of PDR
	[11]	Signal strength of iBeacon	Measures signal strength to identify landmarks	Landmarks were used to assist in correcting the cumulative error of position estimation
	[69]	Coarse–fine square code	Image recognition	Captured images using the camera sensor, performed image recognition and coordinated transformation, and calculated the camera’s central position and orientation
	[70]	QR code	Image recognition	Used QR code information to correct PDR position estimates

**Table 3 sensors-24-06956-t003:** Data used in smartphone-based indoor localization methods.

Method Classification	Localization Dimension	Data Sources and Integration Approach
Single-sensor data	2D/3D	Geomagnetic [2,3], Wi-Fi [9,40], Bluetooth [10], Microphone [1]
Multi-sensor data fusion	2D	PDR [41]
		PDR+ Wi-Fi [7]
		PDR+ Geomagnetic [82]
		PDR+ Wi-Fi+ Geomagnetic [44]
		PDR+ Bluetooth [45]
		PDR+ Acoustic signal [43]
		Wi-Fi+ Geomagnetic [47,48]
	3D	Barometer + Geomagnetic [8]
		Barometer + Wi-Fi [50]
		Barometer + PDR+ Geomagnetic [20]
		Barometer + PDR + Wi-Fi [52]
Integration of sensor data and spatial context	2D/3D	Maps+ Sensors [57]
		Landmarks+ Sensors [65]
		Image data [71], Spatial model [4], Grid data [77] or Graph Model [79] + Sensors

**Table 4 sensors-24-06956-t004:** Key components of machine learning in smartphone-based indoor localization methods.

AI Techniques	AI Algorithms or Models	Utilization	Dataset (Size)
Machine Learning	Fuzzy C-Means Clustering Algorithm	Clustering the data in the geomagnetic fingerprint database [2]	Geomagnetic vector data points (number: 816)
	Improved KNN Algorithm	Building upon the traditional KNN approach, pedestrian motion constraints are incorporated and the KNN algorithm is weighted using PDR location estimates [6]	RSS and offline signal fingerprint library (number:/)
	HMM/YOLOv5	The localization problem based on a dual-layer feature map of vision and Wi-Fi is transformed into a state sequence estimation problem, followed by the detection of safety exit signs [58]	CSI and visual feature data (number:/)
	Random Forest Algorithm /Improved KNN Algorithm	Classify the localization environment during the fingerprint database establishment phase/compute the final location point [59]	RSS data (number:/)
	Least Squares Support Vector Machine	Classifying seed landmarks [56]	Accelerometer and barometer data (number: 1379)
	Decision Tree	Collaborative localization [62]	Map images features (number: 414,500); synthesized walking data (number: 600)
	Decision Tree	Landmark recognition [67]	Sensor data (number: 1200)
	k-means Clustering Algorithm	Used to group data points into multiple clusters for a better understanding and prediction of data characteristics [72]	Wi-Fi fingerprints (number: 300); image data (number: 50)

**Table 5 sensors-24-06956-t005:** Key components of deep learning in smartphone-based indoor localization methods.

AI Techniques	AI Algorithms or Models	Utilization	Dataset (Size)
Deep Learning	CNN	Identifying user behavior patterns [66]	Triaxial acceleration data (number:/)
		Identify ceiling areas and remove them from the feature database [74]	Image data (number: 930)
		Identifying user behavior patterns [46]	Sensor data (number: 28,476 s)
		Identifying user activities and specific behaviors like ascending or descending stairs and using an elevator [54]	Activity logs dataset (number:/)
	LSTM	Identifying unique patterns from the time series of sensor data that are associated with specific locations [21]	Magnetic field data (number: 15,000); light intensity data (number: 18,000)
		Learning the mapping relationship from signal features to location information [7]	RSSI data (number:/)
	Bi-LSTM	Used to extract features from sensor data sources such as Wi-Fi, barometers, and magnetometers, and to predict floor information [19]	Life dataset (number:/)
	DNN (Deep Neural Network)	Learning the nonlinear correlation between the signal strengths received from different base stations and the user’s location [15]	RSS data (number:/)
	Lightweight DNN	Trained on the measurements from the accelerometer and gyroscope to learn the velocity change vectors, and uses these learned features to estimate the pedestrian’s trajectory [43]	IMU, SLAM data (number:/)
	Domain Adaptation Localization Algorithm	Extracting task-relevant and device-independent Wi-Fi data features through adversarial training, and transferring the learned location information from the source domain to the target domain [9]	RSSI data (number:/)
	LF-DLSTM	Processing local features and learning their time series patterns [22]	RSSI data (number: 81,900, 9000)
	Wavelet-CNN	Identifying six complex human motion states, including walking, jogging, jumping, standing, climbing stairs, and descending stairs [44]	MEMS data (number: 461,450)
	KNN Algorithm/BP Neural Network	Floor discrimination: uses Wi-Fi signal strength as a feature input, and the network outputs the predicted floor information [52]	RSSI data (number:/)

**Table 6 sensors-24-06956-t006:** Real-time performance of smartphone-based indoor localization methods.

Method Classification	Real-Time Performance
Single-sensor data	The algorithm performed localization twice per second, with an average single-point matching time of 7.89 ms [2]. The algorithm’s response time was at the millisecond level [10].
Multi-sensor data fusion	The RNN model output a localization result after five RSSI sampling cycles (i.e., 5 s) once sufficient RSSI data had been collected [7]. The update rate was 20 Hz [43]. The update rates for RSSI and MEMS sensor data were 5–10 Hz and 50 Hz [44]. Compared to non-clustered Wi-Fi fingerprint matching methods, the clustered approach significantly reduced the single-point localization time, with an average reduction of 51%, a maximum reduction of 64%, and a minimum reduction of 36% [53].
Integration of sensor data and spatial context	The time for a single localization operation was approximately 150 ms [58]. When the number of particles was set to 100, the map-matching-based PF algorithm converged, leading to improvements in both accuracy and stability, with a computation time of 0.5596 s, roughly equivalent to the time it took for a pedestrian to take 1–2 steps [61]. The time required for standard map matching was 1.536 s, while for semantic road network map matching it was 1.225 s [65]. The average overall localization latency was between 2 and 3 s [70]. The average computation time per walking step was 0.166 s, with the first EKF taking 0.115 s and the second EKF taking 0.051 s [67]. The average computation time was 0.42 s [71]. The average localization time was 0.6 s [72]. Without an initial position, the time required for localization was approximately 6.1 s per image. By incorporating coarse localization results and limiting the image database to a 5 m range, the localization time was reduced to 1.6 s per image. After excluding images captured by overhead cameras, the localization time further decreased to 0.8 s per image [74].

## Data Availability

The data presented in this study are available in this article.

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
