# Peer review of "A Review of Indoor Localization Methods Leveraging Smartphone Sensors and Spatial Context"

_sensors, 2024, doi:10.3390/s24216956_

Round 1
Reviewer 1 Report
Comments and Suggestions for Authors
1. It is proposed to strengthen the description of smartphone indoor positioning application requirements in the introduction. Demand pulls theoretical research, and application pushes technical progress. This is also an important reference for evaluating the advantages and disadvantages of positioning technology.
2. It is suggested to add a section to organize the smartphone sensors, and compare them respectively in terms of observation, cost, availability and reliability.
3. It is suggested to strengthen the organization of algorithm theory rather than technical methods. For example, in multi-sensor integrated localization, data fusion algorithm is the core, and different fusion algorithms will get different results for the same observations. This is also more valuable to the readers.
4. Artificial intelligence algorithm is one of the hotspots of current research, and the discussion in this direction should be strengthened.
5. Spatial context assisted localization is an effective supplement to sensor data, the mainstream of smartphone localization is still based on sensor data, and it is suggested to compress the content of section IV.
Reviewer 2 Report
Comments and Suggestions for Authors
The paper presents a survey of classic and recent advances of indoor localization. In particular, it emphasizes on the applicable method on the smartphone platform. Localization schemes are categorized into single source methods, multi-source fusion methods, with-spatial-context methods.
Overall, the paper is well-structured and easy to follow. Some minor comments are listed as follows.
- In Section 2, the author claims two subcategories based on whether the scheme rely on external signal sources. Magnetic methods are categorized as no external signals. In my opinion, magnetic field is still an external signal which is generated by the earth. It would need to revise the category criteria in the section to make it more accurate.
- In Table 3, methods like Wi-Fi and magnetic field are categorized as 2D schemes. They can indeed work in 3D scenarios as well. Considering a Wi-Fi fingerprint method which contains fingerprints in multiple floors. Even there is no barometer, it can still output the location in different height (level).
- Some Wi-Fi and magnetic schemes are not included. For example,
H. Wu, Z. Mo, J. Tan, S. He and S.-H. Chan, "Efficient Indoor Localization Based on Geomagnetism," ACM Transactions on Sensor Networks, Vol. 15, No. 4, pp. 42:1-42:25, August 2019.
Round 2
Reviewer 1 Report
Comments and Suggestions for Authors
Thanks to the authors for their careful revision. This manuscript is very well to be accepted.